# Large Language Model Guided Tree-of-Thought

## Abstract

In this paper, we introduce the Tree-of-Thought (ToT) framework, a novel approach aimed at improving the problem-solving capabilities of auto-regressive large language models (LLMs). The ToT technique is inspired by the human mind's approach for solving complex reasoning tasks through trial and error. In this process, the human mind explores the solution space through a tree-like thought process, allowing for backtracking when necessary. To implement ToT as a software system, we augment an LLM with additional modules including a prompter agent, a checker module, a memory module, and a ToT controller. In order to solve a given problem, these modules engage in a multi-round conversation with the LLM. The memory module records the conversation and state history of the problem solving process, which allows the system to backtrack to the previous steps of the thought-process and explore other directions from there. To verify the effectiveness of the proposed technique, we implemented a ToT-based solver for the Sudoku Puzzle. Experimental results show that the ToT framework can significantly increase the success rate of Sudoku puzzle solving. Our implementation of the ToT-based Sudoku solver is available on GitHub [1].

## 1 Introduction

Self-attention based auto-regressive large language models (LLMs) such as GPT-4 have recently taken the world by storm [1, 2, 3, 4, 5, 6]. These LLMs excel at a variety of tasks that previously thought as extremely difficult or even impossible. For example, they are able to handle various logical and mathematical reasoning tasks, particularly those that entail "short-range reasonings" necessitating only a few steps to arrive at conclusions [6, 7]. Such remarkable capabilities have even led to speculation that an early form of artificial general intelligence (AGI) may have already emerged [7]. However, today's LLMs still exhibit limitations in certain domains, especially for "long-range" reasoning tasks, where long-term planning and solution exploration are necessary [7]. When presenting a LLMs such as GPT-4 with a challenging problem solving task, especially the so called System-2 reasoning problems [8], the model does not always succeed. Although the generated answer may be indicative of the correct direction, the derivation process frequently includes logical errors. We hypothesize that there are two main contributing factors which limits the problem solving ability of LLMs:

**Lack of correctness checking**: To ensure correctness, a good practice for a human solver is to carry out verification procedures at *every step* of the problem-solving process, thereby ensuring the credibility of the final solution. In comparison, auto-regressive language models do not explicitly perform logical correctness checks as it generates a new token based on the previous tokens. This limits the model's capacity to rectify its own mistakes. A minor error could be amplified as the model generates more tokens, thereby leading to rapid solution quality deterioration and making it difficult to recover from mistakes.

**Solution generated linearly**: As mentioned above, LLMs typically generate a token based on the preceding sequence of tokens without backward editing. On the contrary, when a human solver

---

[1]GitHub link hidden for double-blind review.

Preprint. Under review.

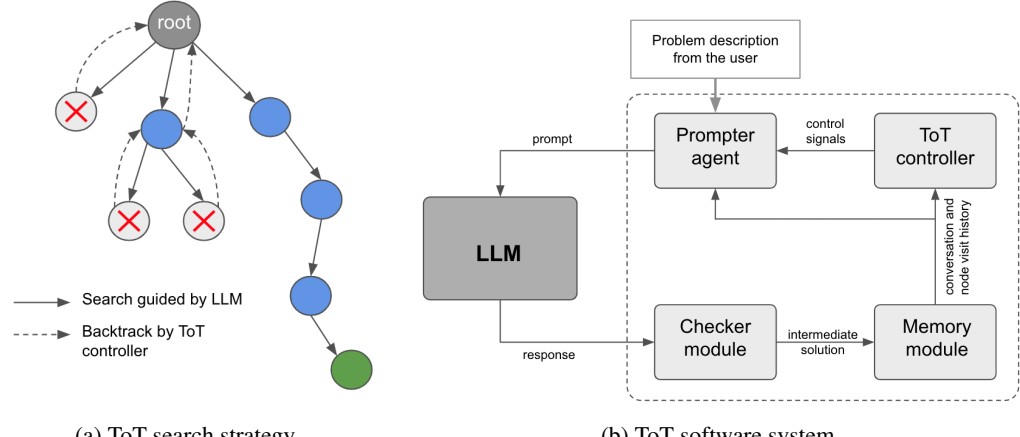

(a) ToT search strategy.  (b) ToT software system.

Figure 1: (a) Details of the Tree-of-Thought search strategy, where a solid arrow means a search step guided by the response from the LLM, and a dashed arrow indicates backtracking commanded by the ToT controller. (b) The software system implementing the Tree-of-Thought search strategy. It enhances the problem solving capability of an LLM by augmenting it with additional modules including a prompter agent, a checker module, a memory module, and a ToT controller.

attempts to solve a problem, she might backtrack to previous steps if a derivation step is incorrect, or if she becomes stuck and is unable to make further progress towards arriving at the final answer. Fields Medal winner Terence Tao once shared his experiences solving hard math problems[2]: "When I was a kid, I had a romanticized notion of mathematics, that hard problems were solved in Eureka moments of inspiration... With me, it's always, Let's try this. That gets me part of the way, or that doesn't work. Now let's try this. Oh, there's a little shortcut here... You work on it long enough and you happen to make progress towards a hard problem by a back door at some point. At the end, it's usually, oh, I've solved the problem." The problem solving process as he described is a *tree-like* thinking process, rather than a *linear* chain-of-thought [9]. The limitation of linear response generation is also apparent from a computational complexity perspective. The number of computation steps an auto-regressive LLM can perform is polynomial in terms of its input length. Unless **P = NP** holds which contradicts the widely accepted belief, there would be problems in **NP** that is not solvable by auto-regressive LLMs.

Inspired by these two shortcomings of auto-regressive LLMs, we propose a novel framework which augments an LLM with several additional modules including an automatic "prompter agent". This framework employs a solution search strategy we call the ***Tree-of-Thought (ToT***[3]***)***. This strategy solves a problem through a multi-round conversation between the LLM and the prompter agent. Figure 1a provides a visual description of the ToT search strategy, in which the LLM plays a crucial role in *guiding* the search for solutions. To make it more concrete, let us assume the problem to be solved is an instance of the Sudoku puzzle. The "root" node represents the initial state, corresponding to when a human mind just reads through the problem description, and begins the thinking process. A blue node in the figure represents a valid partial solution, which can be used by the LLM as a basis to generate the next search step. In the context of Sudoku puzzle solving, this means presenting a partially filled Sudoku board to an LLM and letting the LLM fill in a few more cells. The rationale is that an LLM like GPT-4 has been trained on a vast amount of text corpus which includes many Sudoku puzzle solutions. Given a partially filled board, likely the LLM is able to recognize the pattern, and provide useful insights on how to proceed following the Sudoku rules. Hence, it is highly probable that a search guided by the LLM is significantly more efficient than a brute-force search. In the figure, the search steps guided by the LLM are represented by the solid arrows. However, these steps generated by the LLM are not guaranteed to be always logically correct. Thus, we introduce a "checker module" to perform correctness checks. In Figure 1a, a gray node with an "X" marker represents a "dead-end", i.e. a partial solution that the checker module considers as invalid. For

---

[2]https://newsroom.ucla.edu/releases/Terence-Tao-Mozart-of-Math-7252

[3]The word "tot" means a very young child, which is an interesting analogy as this work is a preliminary exploration into the potential for automated problem-solving utilizing language models.

Sudoku, this means the partially filled board violates the Sudoku rules. If the current node is invalid, obviously we need to return to a parent or an ancestor node in order to correct the mistake. This can be coordinated by a module called the "ToT controller" which oversees the ToT search. With the backtracking capability, the system can regenerate the solution and thus recover from errors. In addition, even when the current node is valid, if the system remains stuck at it for too long, the ToT controller could issue a backtrack signal to explore other possible solutions. This is similar to a scenario where a human mind realizes that there is no viable path towards reaching the final solution through a particular direction, prompting her to change course and explore alternative routes. This process continues until either a full solution is found (represented by a green node in the figure), or a pre-specified maximum round of conversations is reached.

Note that while the above discussion utilized Sudoku solving as a tangible example to illustrate our main ideas, the ToT framework can potentially be applied to more general mathematical and logical reasoning tasks. For example, in the context of mathematical theorem proving, a full solution corresponds to the complete proof, encompassing a total of $n$ derivation steps. On the other hand, a partial solution refers to a subset of these steps, specifically the initial $k$ steps, where $k$ is less than $n$. The checker verifies the logically correctness of a given partial proof. In parallel, the prompter agent and the ToT controller can offer hints and suggestions to the LLM, encouraging it to think about the subsequent proving step, or explore different directions for theorem proving when necessary.

To evaluate the effectiveness of the ToT framework, we implemented a ToT-based Sudoku puzzle solver and evaluate it on a suite of Sudoku puzzle benchmarks we created. As shown by the experimental results in Section 4.2, the ToT framework can significantly increase the success rate of Sudoku puzzle solving.

The remainder of the paper is organized as follows. Section 2 reviews the related literature and compared our approach with the most relevant works. Section 3 provides the details of the ToT system architecture. Section 4 describes our implementation of a ToT-based Sudoku puzzle solver and presents the experimental results. Finally, Section 5 discusses the limitation of the present work, and potential future extensions of the ToT framework.

## 2 Related Works

Developing intelligent systems that can reason has long been one of the primary goals of artificial intelligence [10, 11, 12]. Recent advancements in large language models, particularly the discovery of their emergent properties and in-context learning abilities, have opened up a new avenue for machine reasoning [6, 7, 9]. It is discovered that prompting language models using chain-of-thought and other hints can elicit them to output step-by-step solutions for mathematical and logical reasoning tasks [9, 13]. Building on these findings, recent studies have also explored the practice of sampling multiple solutions and using self-consistency or complexity-based criteria to determine the optimal response [14, 15]. Experiments were also conducted to evaluate the performance of different prompts [15]. The self-taught reasoner (STaR) [16] is a technique which asks an LLM to generate reasoning chains and drop those producing incorrect answers. Then, the model is fine-tuned with the remaining valid reasoning chains.

Despite showing high potential, these techniques often necessitate human involvement. For example, chain-of-thought style prompting techniques require carefully hand-crafted examples and is thus difficult to scale. Consequently, researchers have started to explore the possibility of automatic prompt generation. Early exploration in this domain includes AutoPrompt [17], prefix-tuning [18], and parameter-efficient prompt tuning [19]. This research direction received even more attention lately. In a recent study [20], the authors experimented with training verifiers to check if the solution provided by an LLM to an given mathematical problem is logically correct. If the trained verifier can effectively judge the LLM outputs, it would provide another avenue for prompt evaluation. Automatic prompt engineer [21] examines a method to select the best prompt from a set of model-generated candidates. The three-phase augment-prune-select method was suggested in [22]. It first generates multiple chain-of-thought candidates, which was then pruned based on whether the derived answer matches with the ground truths. Finally, a policy gradient based method was used to select the optimal combination of several rationale chains from the pool for CoT prompting.

Very recently researchers have also turned their attention to augmenting LLM with additional agents for various purposes. This is also the research field that is most relevant to our current work. AutoGPT

[23] is a program which combines GPT-4 with additional modules including an execution agent and a memory unit. It can chain together LLM "thoughts", in order to autonomously achieve whatever goal the user sets. PromptPG [24] proposes an approach that can learn to select in-context examples from a small amount of training data via policy gradient for prompt learning. The PromptPG agent learns to find optimal in-context examples from a candidate pool, with the goal of maximizing the prediction rewards on given training examples when interacting with the GPT-3 environment. DEPS [25] is a proposal that utilizes multi-step reasoning and sub-task error correction to tackle complex tasks with long-range dependencies. By being able to provide explanations for errors in sub-tasks within a trial, DEPS exhibits remarkable performance. ReAct [26] is an approach that employs emergent properties present in LLMs, such as traces of verbal reasoning, to enable agents to reason and take action, resulting in impressive performance on different text-based benchmarks. Building on top of ReAct, Reflexion [27] is an approach that equips an agent with dynamic memory and self-reflection capabilities, improving its existing reasoning trace and ability to choose task-specific actions. To achieve complete automation, a simple but effective heuristic was designed to enable the agent to identify hallucination instances and prevent repetitive action sequences. Our proposal shares some commonalities with these approaches, for example, the use of a memory module and additional agents for automatic prompt generation. However, our approach is unique in that it introduces a ToT controller which can explicitly conduct backtracking when necessary. This not only allows the system to recover from mistakes, but potentially can also enlarge the solution search space.

## 3 Architecture

### 3.1 The Tree-of-Thought Framework

Figure 1b depicts the software system that implements the ToT Framework. As mentioned earlier, it incorporates several components which enhance the problem solving capability of the LLM, including a *prompter agent*, a *checker module*, a *memory module*, and a *ToT controller*.

The problem solving process starts with the user inputting the problem description. The prompter agent then relays the problem to the LLM, with additional prompt text which encourages the LLM to come up with an intermediate solution instead of trying to reach the full solution in a single shot. After receiving the response from the LLM, the checker module is invoked to check the validity of the intermediate solution generated. If it passes the correctness check, the intermediate solution will be parsed and stored in the memory module. Then, based on the content of the memory module, the prompter agent generates a prompt to encourage the LLM to generate the next step. Conversely, if the LLM generates an invalid intermediate solution, the ToT controller will activate the prompter to offer hints to the LLM and request it to consider again. Note that in general, a valid intermediate solution does not always leads to the correct final solution. In order to prevent getting stuck, the ToT controller constantly monitors the search process and determines whether to continue trying from the current node or backtrack to a parent or an ancestor node and explore alternative directions.

The ToT strategy can be viewed as a tree-search algorithm using an LLM as a heuristic for generating the search steps. In this setting, the LLM is only used for the "short-range reasoning" tasks, i.e deriving the next intermediate solution, which is a type of tasks that have been shown to have a high success rate for LLMs [7]. On the other hand, by introducing the checker, the system have a higher likelihood to discover the mistakes it makes as it generates the solutions. Moreover, by allowing the system to backtrack from a valid but somewhat "hopeless" intermediate solution, the system is able to explore a larger solution space, which enhances the "long-range reasoning" capability of the system as a whole. The ToT framework thus combines the best of both world. Furthermore, this multi-round conversation technique increases the number of computation steps the system can perform. Thus, based on the time hierarchy theorem in computational complexity theory [28], the ToT framework can expand the range of problems that can potentially be solved compared to relying solely on a single round of conversation with an LLM.

### 3.2 ToT Modules

In this section we provide more details of the components of the ToT software system.

**Checker Module**. The checker module can either be rule-based or implemented as a deep neural network. For problems that have an explicit polynomial time algorithm for correctness checking (i.e.

problems in **NP**), rule-based checkers can be implemented. Numerous important mathematical and logical problems are in this category, for example, equation solving, polynomial factoring, 3SAT, and puzzles like Sudoku. With a rule-based checker, the ToT software can be viewed as a hybrid system which allows explicitly encoding prior knowledge (e.g. the Sudoku rules) into a neural network powered system. An alternative is to train and use a neural network based classifier as the checker [20]. This is especially useful for problems where a rule-based checker is difficult to implement, e.g. checking whether a mathematical proof is correct.

**Memory Module**. The memory module can be used to store the entire conversation history between the LLM and the prompter agent, as well as other supplemental data useful for problem solving. The data stored can be served as the information source for the prompter agent to generate helpful hints for the LLM.

**ToT Controller**. The ToT controller oversees the entire ToT search. It can be implemented in a number of ways. It can be as simple as encoding two rules: 1) if the checker thinks the current partial solution is invalid, backtrack to the parent node, and 2) if the current partial solution is valid, but the ToT search tree has explored its $C$ children and yet failed to find the final solution, then backtrack to the parent node. Here $C$ is an pre-configured integer.

A more advanced version of the ToT controller can employ a policy network to determine the backtracking policy. The network's inputs include the recent search history comprised of the sequence of the last $k+1$ node visited in the search tree $s_{i-k}, .., s_{i-1}, s_i$ ($k$ is a hyper-parameter). The network also takes in $c_i$, a Boolean variable which indicates whether the checker module considers the current node $s_i$ is valid. We can sample from the policy to determine the next action $a_i$:

$$a_i \sim \pi_\rho^t(a|c_i, s_i, .., s_{i-k}), \ a \in A_{cand} \tag{1}$$

where $\pi_\rho^t$ represents the policy network of the ToT controller with parameters $\rho$. The set of candidate actions $A_{cand}$ includes simply staying at the current node to generate the next step, and backtracking to the parent or an ancestor node at most $L$ levels up in the search tree where $L$ is a hyper-parameter. Thus, we can use one-hot encoding for the actions, where backtracking $j$ levels up is represented by a vector where only the $j$th position is set to 1. The action vector $a$ and checker output $c_i$ are processed by a feed-forward network (FFN) to for deep features extraction. A linear layer with learnable parameters $\mathbf{W}_1$ and $\mathbf{b}_1$ is added on top of the FFN to map its output to a vector $\mathbf{g}(a, c_i)$. The latest $k+1$ visited nodes are concatenated into a string, and then added with position embedding (PE), and finally inputted into a self-attention model [1]. The idea is that by adding position embedding, the attention model will be able to make decisions based on the sequence of the recent node visits. A linear layer with learnable parameters $\mathbf{W}_2$ and $\mathbf{b}_2$ is added on top of the attention model to transform its output to a vector $\mathbf{g}(s_i, .., s_{i-k})$ whose dimension matches with that of $\mathbf{g}(a, c_i)$. Finally, we calculate the inner-products of these two vectors, and use the softmax function to compute the probability of each action candidate:

$$\begin{aligned} \mathbf{g}(a, c_i) &= \mathbf{W}_1 \cdot \text{FFN}(a, c_i) + \mathbf{b}_1 \\ \mathbf{g}(s_i, .., s_{i-k}) &= \mathbf{W}_2 \cdot \text{Attention}(\text{PE}(s_{i-k}||..||s_{i-1}||s_i)) + \mathbf{b}_2 \\ \pi_\rho^t(a|c_i, s_i, .., s_{i-k}) &= \frac{\exp(\mathbf{g}(a, c_i) \cdot \mathbf{g}(s_i, .., s_{i-k}))}{\sum_{a' \in A_{cand}} \exp(\mathbf{g}(a', c_i) \cdot \mathbf{g}(s_i, .., s_{i-k}))} \end{aligned} \tag{2}$$

In the above formula, "||" is the string concatenation operator. Section 3.3 will discuss the training algorithm for the ToT controller policy network.

**Prompter Agent**. The prompter agent gives hints to the LLM for it to generate the next search step. The most basic hint can be a generic prompt using the following template: $generic\_tmpl$ = "*For the given problem: [problem description], we have come up with a partial solution: [partial solution summary]. Please derive the next step on top of this partial solution, and return the next step in the following JSON format {next_step: <next_step>}*". Note that the template requires the LLM to respond with a structured JSON string. This is a trick to make it easier for the checker to extract the next step from the LLM response. To create an actual prompt from this template, the prompter needs the *[problem description]* and the *[partial solution summary]*, both of which can be queried from the memory module.

Similar to the ToT controller, we can also implement the prompter agent as a policy network, which can generate prompts based on the current partial solution and the conversation history. First we

---

**Algorithm 1** Policy Gradient based Training Algorithm for the ToT System

---

1: **Input:** training set $P_{train}$, num of training epochs $N$
2: **procedure** REINFORCE($P_{train}$, $N$)
3:     randomly initialized the ToT Controller policy $\pi_\rho^t$
4:     randomly initialized the Prompter agent policy $\pi_\theta^p$
5:     **for** $epoch$ = 1, 2, .., $N$ **do**
6:         $\pi_w \leftarrow \pi_\rho^t$ if $epoch$ is even, $\pi_\theta^p$ otherwise   ▷ update the selected policy only, fix the other
7:         **for** $p_i \in P_{train}$ **do**
8:             $r_i \leftarrow reward(\text{ToTSystem}(p_i))$   ▷ attempt to solve problem $p_i$ and obtain reward $r_i$
9:             $w \leftarrow w + \alpha \nabla_w \log \pi_w r_i$
10:        **end for**
11:     **end for**
12: **end procedure**

---

define the prompt template as follows: $prompt\_tmpl = generic\_tmpl \parallel$ "*Here are a few examples: [in-context learning examples]*.", where $\parallel$ is the string concatenation operator. The variable *[in context learning examples]* are in-context learning examples for the problem being solved, which can be picked by the prompter policy network from a set of candidates, similar to the PromptPG approach [24]. The rationale is that given the current and recently attempted intermediate solution, some in-context examples might work better than others as hints for the next step. Given the recently visited node sequence $s_{i-k}, .., s_{i-1}, s_i$, our goal is to select $l$ examples $e_i = \{e_i^1, e_i^2, ..., e_i^l | e_i^j \in E_{cand}\}$ where $E_{cand}$ is a pool of in-context learning example candidates. The examples are selected according on a policy:

$$e_i^j \sim \pi_\theta^p(e|s_i, .., s_{i-k}), \ e_i^j \in E_{cand} \text{ for } j = 1, 2, ..., l \tag{3}$$

where $\pi_\theta^p$ represents the policy network of the prompter agent with parameters $\theta$. With the set of selected examples, the prompter agent generates a prompt from the template: $p_i = prompter(prompt\_tmpl, e_i, s_i)$, which can be fed into the LLM to obtain the next intermediate solution $s_{i+1} = LLM(p_i)$. The neural network architecture for the prompter's policy network is similar to that of the ToT controller. The only difference is that since the in-context examples are expressed in natural language, instead of FFN, we use an attention model to process them:

$$\begin{aligned} \mathbf{h}(e) &= \mathbf{M}_1 \cdot \text{Attention}(\text{PE}(e)) + \mathbf{c}_1 \\ \mathbf{h}(s_i, .., s_{i-k}) &= \mathbf{M}_2 \cdot \text{Attention}(\text{PE}(s_{i-k}||..||s_{i-1}||s_i)) + \mathbf{c}_2 \\ \pi_\theta^p(e|s_i, .., s_{i-k}) &= \frac{\exp(\mathbf{h}(e) \cdot \mathbf{h}(s_i, .., s_{i-k}))}{\sum_{e' \in E_{cand}} \exp(\mathbf{h}(e') \cdot \mathbf{h}(s_i, .., s_{i-k}))} \end{aligned} \tag{4}$$

The prompter policy network can be trained together with the ToT controller using multi-agent reinforcement learning methods. The training algorithm of the prompter's policy network is discussed in Section 3.3.

### 3.3 ToT System Training

In the previous sections, we have described the multi-agent ToT framework. This section dives into how we can train the agents, in particular, the policy networks of the ToT controller and the prompter agent. While there are many multi-agent reinforcement learning algorithms (MARL) proposed in the literature [29, 30, 31], in this work we adopt a relatively simple approach which uses a modified version of the REINFORCE algorithm [32] to train the policy networks of the ToT controller and the prompter agent directly. The more advanced MARL algorithms will be explored in the future.

First, we define a run of the ToT system as the process where a user inputs the problem description, and the ToT system attempts to solve the problem until it thinks the problem is solved, or a pre-specified maximum round of conversations is reached. Next, we define the reward $r$ of a run: if the problem is correctly solved, then $r = +1$. Otherwise, if the system outputs an incorrect solution, or the maximum round of conversations is reached, then $r = -1$.

---
**Algorithm 2** Problem Solving Using the ToT System
---
1: **Input:** problem description from the user $p_{user}$, max num of conversation rounds $K$
2: **procedure** SOLVE($p_{user}$, $K$)
3:      $prompt \leftarrow$ Prompter($p_{user}$)
4:      **for** $round$ = 1, 2, .., $K$ **do**
5:          $response \leftarrow$ LLM($prompt$)
6:          $result \leftarrow$ Checker($response$)
7:          **if** $result$.isValidFinalSolution() **then**
8:              **return** ($result.solution$)
9:          **end if**
10:         memory.store($result$)
11:         $ctrl\_signal \leftarrow$ ToTController(memory)
12:         $prompt \leftarrow$ Prompter(memory, $ctrl\_signal$)
13:     **end for**
14:     **return** ($nil$)
15: **end procedure**
---

The training algorithm is provided in Algorithm 1. The algorithm takes two inputs, the training data set $P_{train}$, and the number of training epochs $N$ (Line 1-2). The two policy networks $\pi_\rho^t(a_i|s_i, .., s_{i-k})$ and $\pi_\theta^p(e_i|s_i, .., s_{i-k})$ are randomly initialized (Line 3-4). We train the two policy networks in turns, i.e. training one network with policy gradient while keeping the other fixed (Line 6). To be more specific, when the current $epoch$ is an even number, we select the ToT controller policy $\pi_\rho^t$, and keep the parameters of the prompter agent fixed. Otherwise, we select the prompter agent policy $\pi_\theta^p$ and fix the ToT controller policy. Next, the algorithm updates the parameters of the selected policy network using the policy gradient method (Line 7-9). For each problem in the training data, we attempt to solve it with a ToT system run. Based on the result, we obtain the reward for that run (Line 8). The entire training algorithm runs for $N$ epochs.

### 3.4  Problem Solving Using the ToT System

After the ToT system is trained, we can use it for inference, i.e. problem solving. Algorithm 2 provides the pseudo code for solving problems using the ToT system. It starts with a user inputting description of the problem (Line 1-2). The prompter module then converts the user input into a prompt (Line 3) using a prompt template for user input, for example: $user\_input\_prompt =$ "*For the given problem: [problem description], please derive the first step, and return the step in the following JSON format {next_step: <next_step>}*".

Next, up to $K$ rounds of conversations with the LLM are conducted for problem solving (Line 4). In each round, the LLM first produces a response for the given prompt (Line 5). Then, the checker analyzes the response, and returns a result (Line 6). The result contains the partial solution extracted from the LLM response, as well as information like whether the checker considers the solution as a valid final solution, a valid intermediate solution, or an invalid partial solution, etc. If the solution is a valid final solution, the algorithm simply returns it (Line 7-9). Otherwise, the result is stored in the memory module (Line 10). Based on the content of the memory module, the ToT controller issues control signals, e.g. backtracking for $l$ levels, to the prompter (Line 11). Finally, based on the control signal, the prompter looks up the relevant information from the memory module, and produce the next prompt for the LLM (Line 12). If no valid final solution is found within $K$ rounds of conversations, the algorithm return $nil$ indicating it fails to solve the problem (Line 14).

## 4  Evaluation

This section provides the evaluation methodology and experimental results for our proposed ToT framework. Our evaluation focuses on the ToT-based solver for the Sudoku problem. At the first glance, Sudoku problems seem to be just brain teasers with little practical importance. However, the generalized Sudoku problem on $n^2 \times n^2$ grids of $n \times n$ blocks is known to be NP-complete [33]. If the ToT framework can solve instances of the generalized Sudoku problem (granted that it might takes an exponential number of rounds of conversations), in principle it can handle many other mathematical

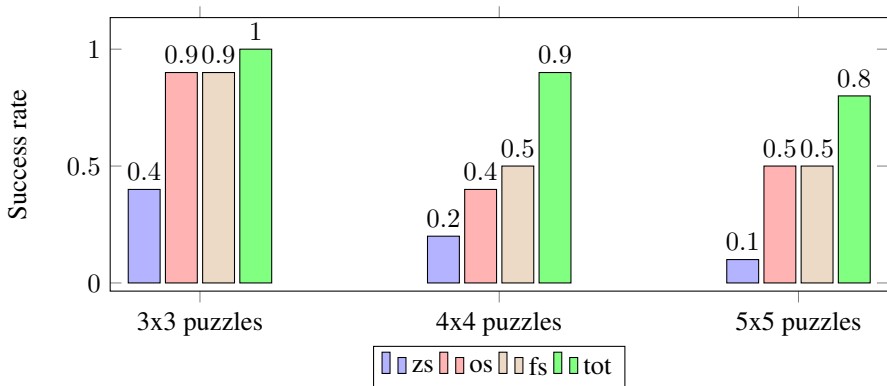

Figure 2: Experimental results comparing the success rate of different LLM-based Sudoku puzzle solvers across three sets of benchmarks.

and logical reasoning tasks. In fact, it is straightforward to re-purpose the implementation described below to solve other puzzles, such as 3SAT, 3-coloring, etc.

Below we first describe the implementation details of the solver. Then, we present the test suite used in our evaluation, as well as the experimental results.

## 4.1 ToT Solver for Sudoku Puzzles

The ToT-based Sudoku solver follows the generic framework described in Section 3 with some specific tweaks for the Sudoku problem. It allows a user to input a Sudoku puzzle using natural languages, for example: "Please solve this 4x4 Sudoku puzzle [[3,*,*,2],[1,*,3,*],[*,1,*,3],[4,*,*,1]] where * represents a cell to be filled".

We have implemented the ToT-based Sudoku solver as described in Section 4.1 in Python. We adopted a rule-based approach for the **checker module** since the Sudoku rules are precise and easy to check. The **memory module** stores the conversation history between the prompter and the LLM, as well as a search tree which maintains all the partially filled Sudoku board the LLM has generated so far. This way, when backtracking happens, the previous board configuration can be retrieved. The **ToT controller** in our implementation is also rule-based. It returns to the parent node in the search tree if either the current node considered invalid by the checker, or the search algorithm has explored more than 5 children of the current node. Finally the **prompter agent** uses a variation of the generic template mentioned above, with the *[problem description]* being the initial configuration of the Sudoku board input by the user, and *[partial solution summary]* being the partially filled board represented by the current node in the search tree. The LLM utilized in this study is the "gpt-3.5-turbo" model, which is accessible through the OpenAI API suite. The *temperature* parameter was set to 1 in our experiments.

## 4.2 Experimental Results

We have implemented four LLM-based Sudoku puzzle solvers and compared their performance: 1) zero-shot solver (**zs**) which directly posts the puzzle description to the LLM, 2) one-shot solver (**os**) which provides a chain-of-thought style step-by-step solution of a 3x3 Sudoku puzzle as an example in addition to the problem description, 3) few-shot solver (**fs**) which provides multiple examples with CoT-style solutions, and 4) our proposed Tree-of-Thought solver (**tot**). We constructed three benchmarks, comprising of ten 3x3, 4x4, and 5x5 Sudoku puzzles, respectively. The objective of a solver is to fill the $n \times n$ Sudoku grid with digits so that each row and column contain all of the digits from 1 to $n$ ($n = 3, 4, 5$ in our experiments).

Figure 2 compares the success rates of different LLM-based solvers across the three benchmarks. Here the term *success rate* refers to the fraction of problems in a benchmark set that are successfully solved by a solver. For example, if a solver is able to solve 4 out of 10 problems in the "3x3 puzzles" benchmark set, then the success rate of this solver for this benchmark set is 0.4. As expected, the zero-shot solver has the worst performance across all the three set of benchmarks. Adding CoT-style

step-by-step examples significantly boosts the success rate, especially for the 3x3 puzzles. This is expected, since one can pretty much rely on "short-range" reasoning skills, which is a strength of the LLM models, to solve a small-sized 3x3 Sudoku puzzle, espcially when CoT-style hints are provided. However, as the puzzle size gets bigger, the success rate of the one-shot and few-shot solver dropped to around 0.5. This is because solving bigger puzzles requires trial and error, which is a capability LLMs generally lack of as discussed earlier.

In comparison, the ToT-based solver demonstrates superior performance when compared to the other solvers. For the 3x3 benchmark set, it was able to solve all the puzzles. The success rate improves by 11% compared to the second best for the two benchmark sets. For the 4x4 benchmark set, the ToT-based solver failed to find the solution for 1 out of the 10 puzzles before reaching the maximum round of conversations (which is set to 100 in our experiments). We suspect it is due to the limited capability of the rule-based ToT controller. In particular, the rule-based controller has no sense of whether the current partially-filled board can be completed without violating the Sudoku rules, which decreases the efficiency of the solution search. We expect a neural network based ToT controller will perform better, which we will verify in the future extension of this work. Despite this, the success rate of the ToT based solver is still 80% higher compared to that of the one-shot and few-shot based solvers. Finally, for the 5x5 puzzles, the ToT-based solver failed with 2 puzzles before reaching the maximum round of conversations. Nonetheless, the success rate is 60% higher compared to that of the one-shot and few-shot based solvers.

## 5 Discussions and Future Works

In this paper, we presented the Tree-of-Thought framework, which enhances LLMs with additional agents and memory modules, resulting in improved performance for mathematical problem-solving tasks. To evaluate the performance of this technique, we implemented a Sudoku puzzle solver based on the ToT framework. One of the limitations of the current implementation is that it utilizes a rule-based checker that contains custom logic, making it less easily adaptable to other problems. For more generic problems, for example, general mathematical and logical reasoning problems, where rule-based solution checking is difficult to implement, a future direction is to explore checkers based on neural network or other probabilistic models. Moreover, the experiments we conducted in this work also uses a rule-based ToT controller, which as we pointed out, has limited capabilities. In the future, we will implement the neural network based ToT controller which can hopefully enhance the system performance. Additionally, the policy-gradient based training algorithm proposed in this work is relatively simple and may be susceptible to training stability issues. To further optimize the ToT system, more advanced multi-agent reinforcement learning algorithms, particularly those designed for cooperative agents, could be adopted.

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
