# OpenReview forum: "Large Language Model Guided Tree-of-Thought"
_NeurIPS.cc/2023/Conference — Submitted to NeurIPS 2023_

### Official Review · Reviewer_gGf6 · 2023-06-12

**Soundness:** 3 good
**Presentation:** 3 good
**Contribution:** 1 poor
**Rating:** 3
**Confidence:** 5

**Summary:**

This paper presents an approach (called Tree-of-Thought, or ToT) for boosting the problem-solving abilities of LLMs by means of backtracking in solution space. The proposed ToT architecture augments the LLM with four modules, and is broadly framed to include multiple potential implementations of those modules, including neural networks (for the prompter and controller) that could be trained by means of policy gradients. One implementation of ToT is evaluated on Sudoku, where it demonstrates significant improvement over an LLM without the ToT augmentations.

**Strengths:**

The motivation is highly topical, as improving the reasoning power of LLMs is a challenge of great research interest and economic value. Most of the paper’s presentation is clear, and the experimental results seem fairly sound, as far as they go.

**Weaknesses:**

The work is limited in four ways.

1 - While the paper lays out a fairly sophisticated and general architecture, only one narrow implementation of ToT is actually tested. Specifically, the current version does not include the policy-gradient options laid out in Algorithm 1 and equations 1-4. Instead, the tested implementation uses the much simpler rule-based controller and LLM-based prompter that have no additional weights to train. This makes it impossible at present to draw conclusions regarding the effectiveness of more advanced realizations of the architecture.

2 - The evaluations, limited as they are to Sudoku, do not demonstrate the generality of the approach, despite the assertion that "in principle it can handle many other mathematical and logical reasoning tasks''. For comparison, consider the recent work of Yao et al., 2023, *Tree of Thoughts: Deliberate Problem Solving with Large Language Models*, which reports evaluations on three separate problem types:  Game of 24, Creative Writing, and 5x5 Crosswords.

3 - Since humans play Sudoku in graphical rather than language-based forms, there is little reason to expect text-only LLMs to perform particularly well on Sudoku at all. And for any algorithmic puzzle (like Sudoku) for which solutions can be explicitly verified, it is unsurprising that explicit checking for valid solutions would boost the performance of an LLM alone. The more central question is how much LLMs contribute to solving such puzzles, but this question is not addressed. The *baselines* tested (zero-shot, one-shot, and few-shot) are actually just *ablations* of ToT, all of them using LLMs.

4 - This paper discusses none of the prior work on Sudoku, such as *SATNet: Bridging deep learning and logical reasoning using a differentiable satisfiability solver* by Wang et al., 2019, which provided a Sudoku benchmark dataset that was used by Ahmed et al., 2013, in *Semantic Strengthening of Neuro-Symbolic Learning*. The *Neural Logic Machine* of Dong et al., 2019, has also been applied to Sudoku (https://github.com/ashutosh1919/neuro-symbolic-sudoku-solver), achieving a 94% success rate on 5x5 puzzles, which is significantly better than the ToT results reported in this paper.

Because of these limitations, the work’s contributions do not seem significant at this point. Nevertheless, the general architecture might prove to be significant once it is fully implemented and tested on a broader set of benchmarks for which baseline results from the literature are available for comparison.

**Questions:**

How many Sudoku instances were tested to produce each bar in Figure 2? Just 10 problems each?

**Limitations:**

Regarding the first limitation described in the Weaknesses section of this review, the paper states "We expect a neural network based ToT controller will perform better, which we will verify in the future extension of this work." However, the crucial point is that policy-gradient training is not evaluated at all, and this limitation is never stated with sufficient clarity. The other three limitations are not adequately addressed.

---

### Official Review · Reviewer_Xo2U · 2023-06-22

**Soundness:** 2 fair
**Presentation:** 2 fair
**Contribution:** 2 fair
**Rating:** 3
**Confidence:** 4

**Summary:**

This paper present tree-of-thought (ToT) as a way of using LLMs to solve problems. ToT involves search + backtracking in a tree-like structure. The work demonstrates the success of this method in simplified sudoku tasks.

**Strengths:**

The idea is interesting, and the Sudoku tasks are a reasonable regime for evaluating it.

**Weaknesses:**

While the method is interesting, the evaluation is minimal. Furthermore, while the sudoku tasks are reasonable as a single setting for evaluation, as the only setting they are quite a narrow domain. The paper also seems to be missing some valuable ablations/baselines. In more detail:

1) The ToT method appears to contain many details which are not individually tested with ablations. Without testing these, it is unclear which aspect is relevant, and what we should learn from the work.
    - For example, the paper emphasizes the benefit of the tree-structured generation, which allows the model to backtrack in its reasoning. However, if I understand correctly, none of the baseline experimental conditions include *any* form of generating multiple answers, or using the checker to verify the answer is correct. For example, rather than a tree search, the model could just run 10 complete rollouts (productions of the correct answer), and then use the checker to identify the correct one (if any). Or, the model could use the checker to decide at each step whether to accept a generation, but without backtracking further. Would one of these methods perform as well? If so, then the key feature is not the tree structure per se, but merely the possibility of generating multiple completions and then checking. In order to understand the contributions of the method, it would be necessary to see these ablations.
    - Likewise, the ToT method appears to benefit from a prompter policy which can decide on the examples for the prompt (if I understand correctly), while the other methods don't; perhaps a prompter is all that's needed.
    - More fundamentally, if I understand correctly (although the paper is not entirely clear on this point), the ToT method is the only one that involves training; there are many ways the one/few-shot prompts could be "trained", such as selecting the examples in the prompt. Ideally, the methods would be evaluated in a setting where the baselines can also benefit from training.

2) The evaluation is quite minimal, in both scope and depth.
    - The method is only tested on a single domain (simplified sudoku, without the box constraints); sudoku is a well-known puzzle, and so is likely to be well-covered in the training corpus. While that does not mean the present results are invalid, it would be useful to see a demonstration of the idea in other, less commmon domains (even other NP ones, such as Traveling salesman, are likely much rarer in the training corpus), or ideally fully novel ones.
    - What evaluation there is only involves a handful of puzzles (10 per condition). The differences between ToT and FS in each condition would not achieve statistical significance by an exact binomial test. While the consistent benefit of ToT across conditions makes it more plausible that there is an overall positive effect, it would be useful to run a larger number of puzzles per condition in order to more accurately assess the magnitude of that effect.

3) The paper could do a more thorough job of situating the present work within the existing literature with similar motivations, e.g. https://arxiv.org/abs/2208.14271 or https://proceedings.neurips.cc/paper/2021/hash/d3e2e8f631bd9336ed25b8162aef8782-Abstract.html

4) The paper presents itself as though it evaluates on Sudoku, but in fact it evaluates on a simplified variant (no boxes). The present abstract, for example, seems somewhat misleading in that it never makes clear that the algorithm is tested on a smaller, simpler version of the task.

5) It would be nice to see comparisons with other language models (especially smaller ones, and ones with an open training process) to understand how general the results are.



**Questions:**

Is there something I misunderstood about the baselines; e.g. are some of them also matched to ToT in terms of training/optimization or compute/calls to the checker?

**Limitations:**

Yes, limitations are discussed.

---

### Official Review · Reviewer_1FWa · 2023-07-03

**Soundness:** 1 poor
**Presentation:** 2 fair
**Contribution:** 2 fair
**Rating:** 2
**Confidence:** 4

**Summary:**

The paper presents a novel algorithm ToT (tree-of-thought) based on:
- LLM (GPT 3.5 in this case)
- checker module (which verifies solutions and partial solutions)
- memory module
- ToT controller that guides the search (it can be a neural network or a set of rules)
- prompter agent (in this paper this is a policy network that selects the best in-context examples for a given tree node

The toT algorithm is tested on a challenging Sudoku task for 3 different board sizes.

**Strengths:**

The approach is novel. It mixes general LLM with two neural networks, trained together. The introduction Section is good: it identifies two main limitations for using LLMs in complex problem-solving. Sudoku is a complex task that requires backtracking and a search, which makes it interesting in the context of ToT. The training of policies is an interesting idea that could be of use in other LLM-based algorithms.

**Weaknesses:**

The biggest weakness of this paper is the small number of experiments, which also are conducted on a single task (sudoku). In the text, many different versions of ToT are discussed, however, experiments are done only for a single setup and a single task. ToT was not tested on any other task, thus we cannot know if it really generalizes at all.

There are far too few experimental results and data. What is missing:
- How many nodes ToT needs on average to solve a given task?
- How many steps baseline needs to solve a given task?
- There are only 3 versions of the Sudoku: for n=3, 4, and 5. (this would be ok if there was more tasks). What happens for n >= 6? If ToT is still the best then it would be great for the method. If not, we will clearly know where is the limit for ToT. If the method is too slow for n>=6 it is important to know.
- What is the price (or number of tokens needed) on average for a single ToT run?
- What was the number of Sudoku puzzles used for evaluation? It is not clearly stated in the text, however, I guess it was 10 boards for each n=3,4,5. If I am wrong please correct me. If it was 10 boards then the results cannot be trusted at all. In such a case for the 0.4 success rate, the 90% confidence interval is (0.15 - 0.7), which tells nothing about the real results. Results on 10 testing boards have no scientific meaning and this is the main reason for such a low score I gave. If I am wrong (and the number is higher I will be happy to improve the score).
-There are no error bars in Figure 2.


Authors claim that: "If the ToT framework can solve instances of the generalized Sudoku [...] in principle it can handle many other mathematical and logical reasoning tasks.". The claim that ToT should be able to handle other complex problems is based on the idea that complex tasks require a similar way of thinking  More advanced mathematical problems like automated theorem proving have their own sources of complexity (e.g. choosing the appropriate lemmas to consider or how to represent the state in a compact form, which fits to the transformer). I know that authors do not claim that for sure ToT works in such tasks, but after reading the paper it seems that the significance of the paper is built upon a promise that ToT can be easily adapted to more serious problems. Since there are no experiments to support this claim I think that the significance of this paper is limited.

There are no experiments concerning other variants of ToT: for example with neural network checker or rule-based policy.

Notation in Algorithm 1 is hard to understand. I had trouble understanding which \pi stands for ToT policy and which for prompt policy. Please consider more natural notation like \pi_{tot} \pi_{prompt} or similar.

Algorithm 2. The meaning of (nil) is not introduced in the algorithm, it is only later in the text.

I think that the version of the text review should use specific LaTeX options: for example line numbers. It is hard for me to refer to concrete lines without them.

It makes no sense for me to describe the procedure of training ToT policy if it was not used in the experiments.

**Questions:**

The most important: what was the number of test boards for each n?

What is the hierarchy theorem? You should briefly explain it in the paper for readers who are not familiar with complexity theory.

Equations (1) and (3): what is s_i precisely? How it is represented?

What networks were trained in the experiments? It is hard to find the text.

**Limitations:**

In the paper, there is no separate section for limitations (some ar ementioned in Section 5). Many missing limitations I already described in the Weaknesses Section of this review.

---

### Official Review · Reviewer_gL3B · 2023-07-08

**Soundness:** 1 poor
**Presentation:** 2 fair
**Contribution:** 2 fair
**Rating:** 4
**Confidence:** 4

**Summary:**

The paper introduces the Tree-of-Thought (ToT) framework, a novel approach to enhance the problem-solving capabilities of large language models (LLMs). The ToT technique mimics the human mind's trial-and-error thought process, allowing LLMs to explore the solution space of complex reasoning tasks and backtrack when necessary. The paper presents an implementation of a ToT-based Sudoku puzzle solver and evaluates its effectiveness on a suite of Sudoku puzzle benchmarks. The experimental results show that the ToT framework can significantly increase the success rate of Sudoku puzzle solving.

The contributions of the paper include introducing a new approach to enhance the problem-solving capabilities of LLMs, presenting an implementation of a ToT-based Sudoku puzzle solver, and demonstrating the effectiveness of the ToT framework on a suite of Sudoku puzzle benchmarks.

**Strengths:**

1. The motivation for moving from linear reasoning, like Chain-of-thought, to a tree-like searching/reasoning is strong and well recognized. Considering the fundamental limitation of autoregressive generation of GPT-like LLMs, we do need more advanced reasoning/search algorithms for better decoding.
2. The proposed method is reasonable and technically sound. The checker module echos the recent findings of self-evaluation of LLMs, and the memory module also is useful in agent-based modeling.
3. The empirical results on the Sudoku pizzles show the effectiveness of the proposed method, esp. when the puzzle becomes harder, the performance of the proposed method is still good.

**Weaknesses:**

1. One of the biggest issues of this paper is the mismatch between the described method and the actual one used in the experiments. The paper spends lots of space talking about how the ToT controller and prompter agent can be modeled by a policy network and trained via multi-agent RL. But it never tried such formulation and training in the experiments and only presented them as some kind of future work. Without valid evidence, empirically or theoretically, the method section is largely questionable.
2. Another issue is the novelty of this work probably is not as big as the paper claims. The formulation of multi-agent RL for controller and agent probably is overcomplicated, and I encourage the authors to think of reformulating them all as LLMs to make things easier. Also, there are many missing related works [1, 2, 3, 4] that have a similar tree search/reasoning formulation with more operational and rigorous experiments.
3. The experiment scope is limited. The proposed method is only demonstrated in one task with the simple formulation of the controller and agent (discussed above). This isn't ok for NeurIPS papers, and we need to better figure out why and how the proposed method can or cannot be applied to more general tasks.

**Questions:**

I raised some questions in the weaknesses section, and there could be lots of improvement space for the authors to make and answer those questions. The following are some minor comments:

1. It seems the catchy name of Tree-of-though has been popularized by another work [5], which draws far more attention than this one; I'd suggest the authors rethink the core contributions of the proposed method and position it in a different and unique way
2. I wonder whether the authors could explicitly explain what kind of search algorithms are used in the proposed method for a better understanding of the method.

[1]. Xie, Yuxi, et al. "Decomposition enhances reasoning via self-evaluation guided decoding." arXiv preprint arXiv:2305.00633 (2023).

[2]. Jung, Jaehun, et al. "Maieutic prompting: Logically consistent reasoning with recursive explanations." arXiv preprint arXiv:2205.11822 (2022).

[3]. Zhu, Xinyu, et al. "Solving math word problem via cooperative reasoning induced language models." arXiv preprint arXiv:2210.16257 (2022).

[4]. Hao, Shibo, et al. "Reasoning with language model is planning with world model." arXiv preprint arXiv:2305.14992 (2023).

[5]. Yao, Shunyu, et al. "Tree of thoughts: Deliberate problem solving with large language models." arXiv preprint arXiv:2305.10601 (2023).

---

### Official Review · Reviewer_QeCS · 2023-07-09

**Soundness:** 2 fair
**Presentation:** 2 fair
**Contribution:** 3 good
**Rating:** 4
**Confidence:** 4

**Summary:**

This paper proposes a tree-of-thought (ToT) framework to improve complex reasoning and problem solving capabilities of auto-regressive language models. Specifically, motivated by how humans process thoughts with trail and error, ToT maintains a memory module, and employs a ToT controller to decide when to proceed on a thought of nodes in a tree, and when to backtrack to a previous parent node depending on a tracker. Evaluated on Sudoku 3x3, 4x4, and 5x5 puzzles, results suggest that ToT is effective compared to few-shot baselines.


**Strengths:**

1. This paper presents an interesting tree of thought method to enable back-tracking in auto-regressive language models. This solves one of the key limitations of LLMs. Similar to humans and inspired by system 2 reasoning, the proposed ToT structure, especially how we can employ a checker to dynamically modify and utilize memory, makes a great contribution to the field, and can inspire future work on how LLMs can be prompted, and even pre-trained.


**Weaknesses:**

1. Although the high-level idea of tree-of-thought is promising, with corresponding ToT controller, agent, and memory, the paper is only evaluated on one Sudoku task, especially when the details of evaluation (e.g., number of games evaluated, and computational cost and prompts used compared to the baselines) are not specified. This makes the evaluation results less convincing. Moreover, despite that the method sounds generalizable, there is no strong evidence on how each module in the framework should be actually implemented to be effective and demonstrated (apart from simply mentioning some future work in the end).
2. It is not clear how each module in the ToT framework should work in details. For example, the memory module seems to be compelling where the LLM can retrieve previous configuration when backtracking, there is no explicit demonstration of how the memory is maintained, and how backtracking would work. Furthermore, the ToT controller is rule-based in the experiment, but Section 3.2 and 3.3 mostly explains how the ToT controller should be trained similar to a policy network. This makes it very confusing to judge the proposed method. I would suggest the authors to add more detailed illustrations using specific examples in the paper revision.

**Questions:**

1. Can you provide more evaluation details for the Sudoku setup? For example, how many games are used for evaluation, and what prompts are employed to the LLM (especially when comparing to the baselines).
2. Why do you use a rule-based controller for backtracking? How do you derive the rules?
3. Can you provide more details in terms of how backtracking interact with the memory?

**Limitations:**

The authors only briefly mentioned the limitations in terms of implantation, but not the limitations of the method overall, such as computational cost.

---

### Author Rebuttal · Authors · 2023-08-06

We sincerely extend our gratitude to the reviewers for their valuable feedback, especially for the thoughtful suggestions regarding the evaluation method, ablation studies, and the discrepancy between the algorithm presented in the paper and its actual implementation for the experimental study. Your input has been very helpful, and we will integrate these suggested changes into the future version of the paper. Appreciate your feedback.

---

> ### Comment · Reviewer_gL3B · 2023-08-10
> **Seperate responses to each reviewer?**
>
> Do we expect separate responses to each reviewer's comments?

---

> > ### Comment · Reviewer_1FWa · 2023-08-11
> >
> > Usually, I would expect, but I am not sure this will happen here.

---

### Decision · Program_Chairs · 2023-09-21

**Decision:**

Reject

**Comment:**

The reviewers appreciate the paper's motivation to improve the problem-solving abilities of large language models (LLMs) using the Tree-of-Thought (ToT) approach, which allows for backtracking in the solution space. The reviewers find the motivation and high-level idea of the ToT framework promising, particularly its potential for addressing the limitations of LLMs and its inspiration from human thought processes.

However, I concur with the reviewers that there are multiple concerns about the limited evaluation of the method and the clarity in the implementation compared to the description . The reviewers provided several suggestions to strengthen the paper's contributions and clarify its novelty.